# Comparison of the Efficiency of Different Eradication Treatments to Minimize the Impacts Caused by the Invasive Tunicate *Styela plicata* in Mussel Aquaculture

**DOI:** 10.3390/ani13091541

**Published:** 2023-05-04

**Authors:** Pedro M. Santos, Eliana Venâncio, Maria Ana Dionísio, Joshua Heumüller, Paula Chainho, Ana Pombo

**Affiliations:** 1MARE—Marine and Environmental Sciences Centre/ARNET—Aquatic Research Network, ESTM, Polytechnic Institute of Leiria, 2520-630 Peniche, Portugal; 2MARE—Marine and Environmental Sciences Centre/ARNET—Aquatic Research Network, Faculdade de Ciências, Universidade de Lisboa, 1740-016 Lisboa, Portugal; 3CINEA and ESTS, IPS—Energy and Environment Research Center, Polytechnic Institute of Setúbal, Estefanilha, 2910-761 Setúbal, Portugal

**Keywords:** non-indigenous species, NIS, invasive species, biofouling, air exposure, freshwater immersion, sodium hypochlorite, acetic acid, mussel farming

## Abstract

**Simple Summary:**

Invasive tunicates have become a global threat to shellfish aquaculture sites in recent decades, particularly mussel farms. In our study, the effectiveness of five eradication treatments (air exposure, freshwater immersion, sodium hypochlorite, hypersaline solution and acetic acid) was tested for the solitary tunicate *Styela plicata.* The effects on blue mussel *Mytilus edulis* survival and growth were also evaluated. The acetic acid treatment was the most effective in eliminating tunicates, although further studies are needed to achieve total survival in mussels. We suggest that the treatments are also likely to produce more effective results as prophylactic measures, applied in controlled environment in mussel seed.

**Abstract:**

In 2017, aquaculture producers of the Albufeira lagoon, Portugal, reported an invasion of tunicates that was disrupting mussel production, particularly the tunicate *Styela plicata* (Lesueur, 1823). A totally effective eradication method still does not exist, particularly for *S. plicata*, and the effects of the eradication treatments on bivalves’ performance are also poorly understood. Our study examined the effectiveness of eradication treatments using three laboratory trials and five treatments (air exposure, freshwater immersion, sodium hypochlorite, hypersaline solution and acetic acid) for *S. plicata*, as well as their effects on survival and growth of blue mussel *Mytilus edulis* Linnaeus, 1758. While air exposure and freshwater immersion caused a 27% mortality rate in *S. plicata*, the acetic acid treatment was the most effective in eliminating this species (>90% mortality). However, a 33–40% mortality rate was registered in mussels. Both species were not affected by the hypersaline treatment in the last trial, but the sodium hypochlorite treatment led to a 57% mortality rate in mussels. Differences in mussels’ growth rates were not detected. These trials represent a step forward in responding to the needs of aquaculture producers. However, further studies are needed to investigate the susceptibility of tunicates to treatments according to sexual maturation, as well as to ensure minimum mussel mortality in the most effective treatments, and to better understand the effects on mussel physiological performance in the long-term.

## 1. Introduction

The growth of the human population leading into the middle of the 21st century poses significant challenges to the supply of high-quality, nutrient-rich food, such as edible marine bivalve mollusks, which are mainly supplied by the aquaculture industry [1]. Mussel aquaculture production has increased globally over recent decades as a result of a decline in wild captures since the early 1990s. In particular, cultured sea mussels (Mytilidae) represent 6% of total mollusk production, reaching 1108 thousand tons in 2020 [1]. Blue mussels (genus *Mytilus*) are cultured globally, particularly in China, the EU and Chile [1,2]. However, despite a 4% increase in value, production has generally declined 26% over the last two decades [1,2,3,4]. Mussel farming is still a slow-growing sector, with low product value and innovation, mainly performed in small-scale semiculture systems, with basic production technology, using longline, raft, bottom or ‘bouchot’ culture techniques [2,5]. In this context, several factors have contributed to the decline in mussel production, including algal blooms, predation, diseases and biofouling [2,6].

Biofouling is the settlement and development of sessile species of microorganisms, plants, algae or animals, known as epibionts, on natural and artificial surfaces [7,8]. The equipment used in mussel farms (e.g., ropes, floats and other infrastructures), in combination with the cultivated mussel shells themselves, represent a favorable habitat that provides substrate, refuge and food, ideal for biofouling settlement [8,9]. Biofouling may result in negative impacts on bivalve cultivation: through increased weight on culture equipment, hampering the harvest process, damage, erosion and altered functioning of the shell, seed losses, competition for space, oxygen and food, and reduced survival and growth. Ultimately, all these factors contribute to a lower bivalve marketability, biosecurity and yield [6,7,10,11].

Fouling communities of shellfish aquacultures are commonly dominated by tunicate species, which may reach high densities in aquaculture sites [12,13,14]. The solitary hermaphroditic tunicate *Styela plicata* (Lesueur, 1823), native to the NW Pacific Ocean, is commonly found in shallow and protected habitats including estuarine areas, most frequently attached to artificial structures, in warm and temperate waters [12,15,16]. With a cosmopolitan distribution, it was introduced into several coastal regions worldwide [12,17,18] and its biological and ecological traits made its expansion successful, including resilience to stressful abiotic conditions [19]. As a pioneer fouling species, *S. plicata* has become a global threat to shellfish aquaculture, over recent decades, particularly mussel farms [12,20,21,22,23,24]. In the Albufeira lagoon, Portugal, mussel farmers have struggled in recent years with an increasing invasion of tunicates, particularly *S. plicata*. Besides prevention strategies for management of tunicates invasions, the development of effective eradication methods is mandatory for introduced populations, despite the associated financial and technical constraints [25,26]. In both research and commercial contexts, physical and chemical eradication techniques have been tested for adult individuals of the genus *Styela*: these include hand removal [27], air exposure [28,29,30], freshwater immersion [30,31], heated seawater [32], acetic acid [26,29,31,32], citric acid [32], sodium hypochlorite [31] and hydrated lime [26]. However, while most of these reports specifically address *Styela clava* (Herdman, 1881), the industry still lacks a totally effective and practical method to effectively address biofouling [7]. Furthermore, current eradication treatments are often detrimental to the bivalves (e.g., induce significant mortality rates); thus, there is a need for further study [7,33]. Our study examined the effectiveness of five eradication treatments on the tunicate *S. plicata*, as well as their effects on survival and growth of the blue mussel *Mytilus edulis* Linnaeus, 1758. We also evaluated gametogenic development of the tunicates to assess a possible association between sexual maturation and susceptibility to treatments.

## 2. Materials and Methods

### 2.1. Animal Collection and Maintenance

At three periods (June, September and November 2021), we manually collected individuals of the tunicate *S. plicata* and the mussel *M. edulis* from the ropes of a mussel raft culture located in the Albufeira lagoon (Sesimbra, Portugal). Animals were transported to the MARE—the Marine and Environmental Sciences Centre (Polytechnic of Leiria)—in isothermal boxes. In the laboratory, individuals of both species were carefully selected and cleaned before an acclimation period of three days in recirculating aquaculture systems (RAS). No mortality was registered during this period in the three trials. Tunicates and mussels were kept in 50 L conical tanks, suspended in fishing nets to simulate the aquaculture environment and supplied with constant aeration. Individuals were exposed to a simulated natural photoperiod. During the trials, temperature, pH, salinity and dissolved oxygen (DO) were measured every two days with a YSI Professional Plus multiparameter probe (YSI Inc., Yellow Springs, OH, USA). Ammonia, nitrite and nitrate were monitored every two days with API^®^ Test Kits (Mars Fishcare, Inc., Chalfont, Pennsylvania, United States of America). The natural seawater was previously filtered through a sand filter and treated with UV light, and a total water exchange was performed daily in all trials. Animals were fed daily with a mixture of live (*Dunaliella tertiolecta* and *Chaetoceros calcitrans*) and frozen (*Tetraselmis chuii*) microalgae, at a concentration of ~100,000 cells/day/animal [34].

Before the application of each treatment and in the end of the trials, tunicates and mussels were measured with a Vernier Calliper (Insize, code 1205-150S, INSIZE Co., Ltd., Zamudio, Spain; ±0.05 mm accuracy) and weighed using an electronic precision balance (Kern PCB 2500-2, Kern & Sohn GmbH, Balingen, Germany; ±0.01 g accuracy). Survival of both species was monitored daily and was assessed as described by Sievers et al. [32] and Cahill et al. [33], for tunicates (siphons closure and response to touch) and mussels (valves closure), respectively. Dead individuals were immediately removed.

### 2.2. Experimental Treatments

#### 2.2.1. Trial 1

A total of 90 *S. plicata* individuals (4.10 ± 0.91 cm length; 2.11 ± 0.46 cm width; 10.27 ± 5.59 g total weight) and 135 mussels (6.79 ± 0.67 cm length; 3.47 ± 0.34 cm width; 29.61 ± 8.92 g total weight) were selected for trial 1. Two experimental treatments for tunicate eradication were tested (air exposure and freshwater immersion) (Table 1). This trial was run in triplicate and included a control group. After the initial exposures, animals were reared for 30 days. For the air exposure treatment, tunicates and mussels were submitted to 6 h of air exposure at the beginning of the trial (T0). The same process was repeated once after 15 days of rearing. For the freshwater immersion, animals were submerged in freshwater for 30 min (T0) and 15 days later, the treatment was repeated for 1 h. Animals were uniformly distributed in 9 tanks, with 10 tunicates and 15 mussels per tank, in a total of 30 and 45 individuals per treatment, respectively. During the trial, the parameters were the following: 20.1 ± 0.6 °C, 8.1 ± 0.3 (pH), 32.6 ± 0.4 (salinity) and 91 ± 1% (DO). The room temperature was 20 ± 1 °C. The biometric parameters of both species are represented in Table 2. Mussels’ specific growth rate (SGR) was calculated as follows:SGR (% day^−1^) = [(ln (M_final_) − ln (M_initial_))/*t*] × 100(1)
in which M_final_ and M_initial_ represent the final and initial average mass (g) of the individuals, respectively, in each replicate tank and *t* represents the number of days.

#### 2.2.2. Trial 2

A total of 180 tunicates (5.62 ± 1.09 cm length; 3.23 ± 0.55 cm width; 29.22 ± 12.05 g total weight) and 180 mussels (5.42 ± 0.59 cm length; 2.91 ± 0.29 cm width; 15.48 ± 4.59 g total weight) were selected for trial 2. Three experimental treatments for tunicate eradication were tested (acetic acid, sodium hypochlorite, hypersaline solution) (Table 1). This trial was run in triplicate and included a control group. For the acetic acid (AcOH) treatment, organisms were submerged in a 4% AcOH solution for 1 min using glacial acetic acid (ACS grade) (Carlo Erba, Val de Reuil, France) (adapted from Sievers et al. [32]). In the sodium hypochlorite (NaClO) treatment, animals were submerged in a 0.5% NaClO solution, prepared with commercial grade bleach (7.5%) for 1.5 min (adapted from McCann et al. [35] and Denny [36]). For the hypersaline group, individuals were submerged in a hypersaline solution (60-salinity) for 20 sec (adapted from Carman et al. [37]), using commercial sea salt (Aquaforest, Poland) to adjust the salinity of the natural seawater. Animals were then uniformly distributed in 12 tanks, with 15 individuals per tank, for a total of 45 individuals per treatment for each species, and maintained for 15 days. The parameters were the following during the trial: 19.6 ± 0.8 °C, 8.0 ± 0.5 (pH), 32.4 ± 0.5 (salinity) and 91 ± 2% (DO). The biometric parameters of both species are represented in Table 3. 

#### 2.2.3. Trial 3

A total of 270 tunicates (4.72 ± 0.75 cm length; 2.66 ± 0.37 cm width; 16.85 ± 5.59 g total weight) and 270 mussels (5.01 ± 0.85 cm length; 2.70 ± 0.36 cm width; 12.73 ± 5.89 g total weight) were selected for trial 3. Based on the results of trial 2, two additional experimental treatments were tested using acetic acid and the hypersaline solution (Table 1). This trial was run in triplicate and included a control group. Two distinct size classes of mussels were selected (lower class: 4.23 ± 0.35 cm length; higher class: 5.79 ± 0.33 cm length) for a total of 6 treatment groups. Animals were uniformly distributed in 18 tanks, with 15 individuals per tank, for a total of 45 individuals per treatment for each species, and maintained for 15 days. The parameters were the following during the trial: 19.1 ± 0.5 °C, 8.2 ± 0.4 (pH), 32.1 ± 0.8 (salinity) and 92 ± 1% (DO). The biometric parameters of both species are represented in Table 4. Tunicates were immediately fixed in a 4% buffered formaldehyde solution for 48 h and stored in 70% ethanol for histological analysis.

The tunicates’ gonadosomatic index (GI) was calculated as follows:GI (%) = (gonads weight/total weight) × 100(2)

### 2.3. Tunicates’ Gametogenic Development

The gonads of tunicates were processed in a Leica^®^ TP1020 Automatic Tissue Processor (Leica Microsystems GmbH, Wetzlar, Germany) with sequential submersions in graded ethanol for dehydration followed by xylene for clarification and impregnation with paraffin wax at 60 °C. After the gonad samples were embedded in 100% (*v*/*v*) paraffin, they were cut with a thickness of 7 µm using an Accu-Cut^®^ SRM™ 200 Rotary Microtome (Sakura Finetek Europe BV, Alphen aan den Rijn, The Netherlands) and stained with Harris’ haematoxylin solution (Scharlab S.L., Sentmenat, Barcelona, Spain) and eosin Y (yellowish) (VWR International, Leuven, Belgium). Gonad tissues were then analyzed using a Leica^®^ DM 2000 LED light optical microscope equipped with a Leica^®^ MC170 5MP HD Microscope Camera and the combined LAS v4.4.0 software (Leica Application Suite) for monitor display (Leica Microsystems GmbH, Wetzlar, Germany). Determined by size and histological characteristics, the oocytes were classified according to their developmental stage into three classes (Sciscioli et al. [38] and Pineda et al. [39]). This allowed us to identify the stage of maturation of each individual as follows:Stage I (pre-vitellogenic)—oocytes smaller than 50 μm, strongly basophilic, with a large nucleus occupying most of the cytoplasm.Stage II (vitellogenic)—oocytes between 50 and 150 μm; the first follicular cells become visible, and the primary follicle begins the process that will give rise to two layers of cells, an outer layer composed of flattened cells and an inner layer accompanying some test cells.Stage III (mature)—oocytes larger than 150 μm; the cytoplasm continues to accumulate nutrient material and increase in volume, accompanying the test cells and the two layers of follicular cells (inner and outer).

For male follicles, a categorical maturity index was established, according to the same authors:Stage I (immature)—male follicles filled only by spermatogonia.Stage II (mature)—male follicles filled with several mature sperm in the lumen.Stage III (spawning)—male follicles with empty spaces in the lumen.

### 2.4. Statistical Analysis

Results were expressed as mean ± standard deviation (SD) and a significance level of α = 0.05 was used for statistical tests. These were performed using IBM SPSS^TM^ Statistics for Windows, version 28 (IBM Corporation, Armonk, NY, USA). The Pearson’s chi-square test was applied to assess a possible association between treatments and organisms’ mortality. Mortality was expressed as mean ± SD of the three replicate tanks by treatment group. To assess differences between treatments, for each biometric parameter (length, width, total weight, mussel SGR), results were analyzed through one-way ANOVA. A Kruskal–Wallis test was used when the data did not meet the assumptions of ANOVA. Data were tested for normal distribution with the Shapiro–Wilk normality test and for homogeneity of variances with the Levene test.

## 3. Results

### 3.1. Trial 1

Both experimental treatments led to a total tunicate mortality of 26.7 ± 11.6%, and 10.0 ± 10.0% in the control (Figure 1). No significant association between treatments and mortality was detected [χ^2^ _(2)_ = 3.34; *p* = 0.19]. A mortality rate of 8.9 ± 3.8% was registered in the mussels control group (Figure 1), showing a significant difference between treatments [χ^2^ _(2)_ = 8.24; *p* = 0.02]. No significant differences were detected for biometric parameters (Table 2) between T0 and T1, for both species, including mussel SGR (Table A1 in Appendix A).

**Table 2 animals-13-01541-t002:** Biometric parameters of the tunicate *Styela plicata* and the mussel *Mytilus edulis* at the beginning (T0) and end (T1) of a 30-day rearing period in which both species were submitted to experimental treatments (air exposure and freshwater immersion). Experiments were run in triplicate with a control group.

	Control	Air Exposure	Freshwater Immersion
T0			
Tunicate length (cm)	4.15 ± 1.01	4.11 ± 0.85	4.06 ± 0.89
Tunicate width (cm)	2.06 ± 0.42	2.13 ± 0.53	2.15 ± 0.44
Tunicate total weight (g)	10.12 ± 5.61	10.44 ± 6.16	10.24 ± 5.13
Mussel length (cm)	6.81 ± 0.70	6.79 ± 0.52	6.76 ± 0.78
Mussel width (cm)	3.46 ± 0.36	3.48 ± 0.36	3.48 ± 0.31
Mussel total weight (g)	29.87 ± 9.95	29.40 ± 8.01	29.56 ± 8.90
T1			
Tunicate length (cm)	3.14 ± 0.69	3.21 ± 0.89	3.13 ± 0.40
Tunicate width (cm)	1.91 ± 0.36	2.02 ± 0.44	2.09 ± 0.38
Tunicate total weight (g)	5.90 ± 2.78	6.35 ± 3.99	6.98 ± 2.89
Mussel length (cm)	6.88 ± 0.70	6.82 ± 0.51	6.90 ± 0.62
Mussel width (cm)	3.53 ± 0.40	3.49 ± 0.34	3.51 ± 0.36
Mussel total weight (g)	30.24 ± 10.41	29.69 ± 8.07	29.84 ± 8.83
Mussel SGR (% day^−1^)	0.04 ± 0.07	0.03 ± 0.00	0.03 ± 0.01

### 3.2. Trial 2

The acetic acid treatment caused the highest tunicate mortality (91.1 ± 7.7%), associated with a 33.3 ± 6.7% mussel mortality (Figure 2). The hypersaline solution promoted a mortality rate of 73.3 ± 6.7% in tunicates, while all mussels survived in this treatment. In the sodium hypochlorite immersion, 66.4 ± 3.9% of the tunicates died, but the highest mussel mortality was also registered (55.6 ± 20.4%). In the control group, a mortality of 57.8 ± 10.2% and 2.2 ± 3.9% was registered in the tunicates and mussels, respectively. A significant association between treatments and mortality, both in tunicates [χ^2^ _(3)_ = 13.87; *p* = 0.00] and mussels [χ^2^ _(3)_ = 54.42; *p* < 0.001], was registered. No significant differences in biometric parameters (Table 3) were found between T0 and T1 for both species, including mussel SGR (Table A2).

**Table 3 animals-13-01541-t003:** Biometric parameters of the tunicate *Styela plicata* and the mussel *Mytilus edulis* at the beginning (T0) and end (T1) of a 15-day rearing period in which both species were previously submitted to experimental treatments (acetic acid, sodium hypochlorite, hypersaline solution). Experiments were run in triplicate with a control group.

	Control	Acetic Acid	Sodium Hypochlorite	Hypersaline Solution
T0				
Tunicate length (cm)	5.72 ± 1.13	5.55 ± 1.12	5.56 ± 1.09	5.60 ± 1.06
Tunicate width (cm)	3.24 ± 0.50	3.19 ± 0.66	3.24 ± 0.53	3.26 ± 0.51
Tunicate total weight (g)	29.91 ± 11.82	28.56 ± 12.23	29.05 ± 11.65	29.34 ± 12.84
Mussel length (cm)	5.44 ± 0.59	5.44 ± 0.52	5.34 ± 0.65	5.45 ± 0.61
Mussel width (cm)	2.93 ± 0.33	2.93 ± 0.27	2.87 ± 0.26	2.93 ± 0.32
Mussel total weight (g)	15.69 ± 4.79	15.45 ± 4.60	15.35 ± 3.44	15.43 ± 5.45
T1				
Tunicate length (cm)	5.48 ± 1.30	4.80 ± 1.26	5.07 ± 1.28	5.19 ± 1.12
Tunicate width (cm)	3.49 ± 0.52	3.03 ± 0.70	3.28 ± 0.75	3.29 ± 0.62
Tunicate total weight (g)	26.65 ± 11.29	21.39 ± 12.17	24.32 ± 10.94	25.91 ± 12.50
Mussel length (cm)	5.48 ± 0.59	5.53 ± 0.57	5.51 ± 0.47	5.45 ± 0.58
Mussel width (cm)	2.88 ± 0.32	2.98 ± 0.29	2.94 ± 0.23	2.92 ± 0.31
Mussel total weight (g)	16.11 ± 5.00	16.43 ± 5.28	16.31 ± 3.64	15.91 ± 5.79
Mussel SGR (% day^−1^)	0.18 ± 0.12	0.42 ± 0.18	0.37 ± 0.16	0.21 ± 0.16

### 3.3. Trial 3

As shown in Figure 3, the mortality rate of tunicates in the lower size class of the mussel trial was the following: 4.4 ± 7.7% (control); 97.8 ± 3.8% (acetic acid); 2.2 ± 3.8% (hypersaline). Within the mussel higher size class groups, the mortality of tunicates was the following: 0.0 ± 0.0% (control); 97.8 ± 3.8% (acetic acid); 2.2 ± 3.8% (hypersaline). A significant association was detected between treatments and the mortality of tunicates in both cases [χ^2^ _(2)_ = 117.96; *p* < 0.001] [χ^2^ _(2)_ = 126.20; *p* < 0.001], respectively.

The mortality of mussels in the lower size class was as follows: 0.0 ± 0.0% (control); 40.0 ± 6.7% (acetic acid); 2.2 ± 3.8% (hypersaline) (Figure 3). A similar result was also obtained in the larger mussel groups: 0.0 ± 0.0% (control); 35.6 ± 10.2% (acetic acid); 2.2 ± 3.8% (hypersaline). A significant association was detected between treatments and mortality in both cases [χ^2^ _(2)_ = 37.61; *p* < 0.001] [χ^2^ _(2)_ = 32.44; *p* < 0.001], respectively. However, mortality was similar between size classes, either in the acetic acid [χ^2^ _(1)_ = 0.19; *p* = 0.66], or in the hypersaline treatment [χ^2^ _(1)_ = 0.00; *p* = 1.00]. No significant differences in biometric parameters (Table 4) were detected in T0 or T1 for both species (Table A3), including mussel SGR (Figure 4). The maturity index of *S. plicata* is detailed in Table 5.

**Table 4 animals-13-01541-t004:** Biometric parameters of tunicate *Styela plicata* and two size classes of mussel *Mytilus edulis* at the beginning (T0) and end (T1) of a 15-day rearing period in which both species were previously submitted to experimental treatments (acetic acid, hypersaline solution). Experiments were run in triplicate with a control group.

	Mussel Lower Size Class Group			Mussel Higher Size Class Group		
	Control	Acetic Acid	Hypersaline Solution	Control	Acetic Acid	Hypersaline Solution
**T0**						
Tunicate length (cm)	4.70 ± 0.70	4.81 ± 0.76	4.66 ± 0.82	4.63 ± 0.72	4.85 ± 0.80	4.66 ± 0.72
Tunicate width (cm)	2.69 ± 0.43	2.66 ± 0.39	2.71 ± 0.40	2.65 ± 0.35	2.59 ± 0.28	2.63 ± 0.37
Tunicate total weight (g)	16.83 ± 6.04	16.82 ± 4.99	16.93 ± 5.77	16.87 ± 5.79	16.72 ± 5.41	16.92 ± 5.82
Mussel length (cm)	4.22 ± 0.31	4.23 ± 0.33	4.24 ± 0.40	5.77 ± 0.35	5.80 ± 0.33	5.81 ± 0.31
Mussel width (cm)	2.40 ± 0.16	2.37 ± 0.21	2.40 ± 0.18	3.02 ± 0.19	3.04 ± 0.20	2.99 ± 0.15
Mussel total weight (g)	7.18 ± 1.46	7.23 ± 1.53	7.22 ± 1.61	18.19 ± 2.85	18.20 ± 2.02	18.35 ± 2.47
Tunicate GI (%)	4.23 ± 1.00					
Tunicate oocyte diameter (µm)	141.31 ± 50.85					
**T1**						
Tunicate length (cm)	3.50 ± 0.65	3.20	3.55 ± 0.90	3.61 ± 0.57	4.00	3.92 ± 0.71
Tunicate width (cm)	2.63 ± 0.45	2.50	2.57 ± 0.55	2.69 ± 0.36	2.80	2.46 ± 0.44
Tunicate total weight (g)	12.43 ± 4.48	9.31	12.97 ± 5.88	13.14 ± 4.72	15.48	14.20 ± 5.73
Mussel length (cm)	4.32 ± 0.36	4.34 ± 0.28	4.37 ± 0.30	5.83 ± 0.34	5.80 ± 0.30	5.88 ± 0.33
Mussel width (cm)	2.46 ± 0.20	2.45 ± 0.19	2.47 ± 0.17	3.06 ± 0.23	3.10 ± 0.24	3.01 ± 0.16
Mussel total weight (g)	8.03 ± 1.46	7.80 ± 1.43	7.96 ± 1.62	18.35 ± 2.72	18.30 ± 2.41	18.60 ± 2.33
Tunicate GI (%)	1.41 ± 0.78	1.42	1.82 ± 0.61	1.74 ± 0.85	0.91	1.64 ± 0.96
Tunicate oocyte diameter (µm)	86.99 ± 39.93	49.78	93.55 ± 42.32	101.96 ± 44.69	67.63 ± 39.89	95.82 ± 41.51
Mussel SGR (% day^−1^)	0.75 ± 0.12	0.50 ± 0.05	0.65 ± 0.16	0.06 ± 0.07	0.05 ± 0.25	0.09 ± 0.11

## 4. Discussion

### 4.1. Air Exposure and Freshwater Treatments

The eradication treatments for the tunicate *S. plicata* showed distinct levels of effectiveness for tunicate mortality and survival of the mussel *M. edulis.* Air exposure and freshwater immersion only promoted a mean tunicate mortality of 27% at the end of the 30-day rearing, observed mainly after the second exposure to the treatments. Hillock and Costello [28] obtained total mortality of *S. clava* submitted to air exposure for 24 h in full sun ambient (15–29 °C) and 48 h in shade ambient (15–27 °C). Therefore, the efficiency of the 6 h air exposure treatment (~20 °C) that we applied would likely be higher in field conditions but may compromise mussel survival. In mussels, air exposure in increased temperatures promote higher oxidative stress, with a negative impact in physiological performance [40]. A shorter exposure would likely be more effective, and particularly more feasible, if applied to mussel seeds, in a controlled environment. Darbyson et al. [41] showed that a 48 h exposure was not effective against *S. clava*, possibly demonstrating that the probability of survival is dependent on size/age [28] and on the level of aggregation of the organisms. Because a 6 h period is the average period of emersion of mussels during low tide [42], a longer exposure would likely compromise mussel productivity, as demonstrated by LeBlanc et al. [29]. Those authors obtained a 40% biomass reduction of *M. edulis* after 7 months in the field, following a 40 h air exposure at 21 °C.

The relatively low mortality of *S. plicata* in the freshwater treatment may be explained in part by its capacity to adapt to low salinity conditions [43]. Rolheiser et al. [44] reported that a freshwater immersion for 0.5–10 min did not reduce the colonial ascidian *Didemnum vexillum* biofouling, as indeed much longer exposures are needed [35], but induced a slight mortality in oysters. Carver et al. [45] reported a 10% mortality in solitary tunicate *Ciona intestinalis* exposed to freshwater for 1 min at 15 °C. However, they reported 66% mortality at 40 °C. At this temperature, some mussel mortality was registered, perhaps due to the synergistic effects of osmotic and heat stress [46]. A 5 min freshwater spray appeared to be effective in eliminating various species of tunicates attached to oysters grown in commercial operations, but detailed information is missing [20]. Although relatively low salinities promote physiological stress in *M. edulis*, the species show highly efficient acclimatization mechanisms in long- and short-term exposures [46,47]. Furthermore, a pronounced osmotic shock promotes a lasting valve closure in *M. edulis*, which likely served as the main mechanism for resisting hyposalinity in the present trial [47]. Therefore, given the absence of mussel mortality and the high tolerance of solitary ascidians to freshwater immersions, increasing exposure periods to freshwater at higher temperatures (30–40 °C) should be tested in *S. plicata* and *M. edulis* [31]. However, the applicability of the treatment in field conditions should be further investigated. A prophylactic approach with mussel seeds translocated from other locations to the aquaculture facilities would also allow the preventing of the introduction of fouling species, using simpler and shorter freshwater exposures.

### 4.2. Sodium Hypochlorite Treatment

The collection of organisms for trial 2 took place in September 2021, when the lagoon water showed relatively higher levels of eutrophication, due to the closure of the tidal inlet by the end of the month. The potentially stressful abiotic conditions, particularly higher temperatures, as well as the transportation and laboratory acclimation periods, may have affected fitness of *S. plicata*, which was reflected in the mortality registered in the control group. Therefore, the results of tunicate mortality in our study should be analyzed with caution. Sodium hypochlorite (bleach) only eliminated 9% more of *S. plicata* comparatively to the control. Similarly, Carver et al. [45] obtained no mortality of *C. intestinalis* exposed to sodium hypochlorite for a longer period of 20 min. On the contrary, Piola et al. [48] reported a 75–100% removal of fouling biota, including the tunicate *C. intestinalis*, sprayed with 20% bleach (0.5–12 h), but only 0–50% removal with 5 and 10% bleach. In the work of Coutts and Forrest [31], *S. clava* was successfully eliminated in a 6 h exposure to sodium hypochlorite; however, in field trials performed on marina pontoons, treatments were not so effective, mainly due to the rapid decline in free available chlorine in the water. Given the short exposure time applied in our study (1.5 min), this issue probably did not represent a significant factor. Once again, the application of the treatment in earlier stages of mussel production, in more controlled conditions, would result in enhanced efficiency. Nonetheless, those results indicate that the concentration and exposure time applied in our study should have been higher to obtain total tunicate mortality. However, some results that have been reported in the literature are contradictory. McCann et al. [35] obtained total mortality of *D. vexillum* immersed in 1% bleach for 10 min, while immersions of 2 and 5 min only produced an initial decline in the surface area of the colony. However, Denny [36] indicates that bleach concentrations as low as 0.1%, applied for just 2 min, are effective in the same species. Nonetheless, the pronounced morphological differences between the colonial *D. vexillum* and the solitary *S. plicata* should be noted. Furthermore, Denny [36] also only obtained a maximum mortality of 6% in the mussel *Perna canaliculus* exposed to 0.5% bleach for 2 min, while in the present study, 55.6% of *M. edulis* died in the bleach treatment. Hypochlorite potentially causes a toxicological response in mussels, mostly in gills, promoting an oxidation process, but it also can affect the mussels’ byssus gland [36,49]. However, the resilience to sodium hypochlorite varies significantly according to different factors, including mussel size, sexual maturation, acclimatation temperature and species [50]. Therefore, despite the apparent resiliency of *S. plicata* to chlorine [31], further studies are needed to ensure *M. edulis* survival in more efficient treatments. Moreover, although the chemicals used in this study are considered to have a relatively low toxicity, with a high biodegradability [31,48], their application in the field must include a consideration of the safety of the operator, as well as environmental mitigation techniques [45].

### 4.3. Hypersaline Treatment

The hypersaline (brine) treatment was not successful in eliminating *S. plicata* which, according to Sims [43], exhibits some level of hyperosmotic regulation. The treatment only promoted a 16% higher mortality in the tunicates of trial 2, comparatively to the control. Similarly, Carver et al. [45] only obtained 25% mortality in *C. intestinalis* exposed to a more intense treatment of saturated brine for 8 min. Carman et al. [37] eliminated various species of tunicates from socks of juvenile *M. edulis* exposed to 10 or 20 s brine baths (70 salinity), but direct comparisons would also be biased due to the different target species. For *D. vexillum* eradication, brine treatments require at least 4 h immersions [35,44]. Vickerson et al. [51] and Rolheiser et al. [44] reported no mortality in mussels (*Mytilus* spp.) and oysters, respectively, while Carman et al. [37] reported 8–30% *M. edulis* mortality in a 20 s brine bath. In comparison with our study, this higher mortality may be due to the use of juvenile mussels, as well as a higher bath salinity. Therefore, increasing the exposure time to a 60-salinity bath may result in mussel mortality, or at least in higher metabolic activity and altered immune system, which would likely affect production [52]. According to Carman et al. [20], 10 min immersions in brine solution followed by a 2 h air exposure are effective in removing tunicates from oyster aquaculture operations. This procedure could potentially increase tunicate mortality while being safe for mussels. However, the logistical or financial viability of the treatment in field conditions should be further investigated [44].

### 4.4. Acetic Acid Treatment

The acetic acid treatment was the most successful in eliminating *S. plicata.* In trial 3, 98% of tunicates exposed to the treatment died, as well as 36–40% of mussels. Similarly, Coutts and Forrest [31] also achieved total mortality of *S. clava* with an identical procedure. In the work of Sievers et al. [32], 1 min immersions in 2 or 5% acetic acid, at ambient temperature, killed 50% of *S. clava* specimens. Only the 2% acetic acid treatment at 40 °C for 1 min led to total mortality. Compared to our study, the different results may derive from distinct physiological conditions and size of the organisms, experimental environments, as well as physiological differences between species. Furthermore, Sievers et al. [32] only assessed mortality in the first 48 h. Although in the present study most mortality was also detected in the first days of exposure, further mortality was registered during the trial period. Carver et al. [45] reported a 95% mortality of *C. intestinalis* exposed to 5% acetic acid for 15 to 30 s, either by spraying or immersion. Forrest et al. [53] also achieved an 84–100% biomass reduction of fouling organisms, mostly dominated by *C. intestinalis*, using 2 and 4% acetic acid in 1–4 min immersions. Nonetheless, *Styela* species may present a higher resistance to some eradication treatments given their thicker tunic, when compared to the soft-bodied *C. intestinalis* [32]. For *D. vexillum*, a 5% concentration was effective in significantly reducing tunicate coverage using 0.5–10 min exposures [37,44,48]. Tunicate mortality can be further enhanced by an air exposure period after immersions, but it would also enhance the probability of mussel mortality, particularly if the acid residue is not rinsed [33,53]. Moreover, as sexually mature mussels are likely more sensitive to chemical treatments [50], the use of mature *S. plicata* individuals in trial 3 may also have favored higher efficiency of the acetic acid treatment. However, further studies are needed to confirm this possibility and also the long-term effect of treatments on the reproductive performance of surviving individuals. Cahill et al. [33] registered a relatively higher sensitivity of the mussel *P. canaliculus* to acetic acid, compared to the oyster *C. gigas*. The authors recommended a concentration limit of 4% to avoid pronounced mussel mortality, which is in accordance with the present results. The authors tested in the laboratory a provisional treatment of 2% acetic acid for 1 min, which resulted in a higher mussel productivity in the field, as a result of biofouling reduction. Other works confirm these results. LeBlanc et al. [29] reported 67 and 74% reductions in *M. edulis* socks’ weight after exposure to 30 s and 2 min acetic acid immersions, respectively, as a 5% concentration was applied in mussel seed. However, loss of attachment may have also contributed to weight reduction, making it difficult to directly compare results. Carman et al. [37] also reported total juvenile *M. edulis* mortality in the acetic acid treatments (15–25 mm shell length). Although the authors suggested immersion times of less than 5 min in future works, larger mussels exposed to 1 min in the present study still experienced relatively high mortality. In fact, acetic acid concentration is likely more relevant than exposure time [33], although 5–10 s may be insufficient to eliminate tunicates [45]. Sievers et al. [32] also reported mussel (*Mytilus galloprovincialis*) and oyster mortality in the abovementioned treatments for *S. clava*, but it was only significant with the acetic acid at 50 °C. In the study of Rolheiser et al. [44], high mortality occurred in oysters exposed to a concentration of 5%. Apart from mortality, acetic acid may also affect mussel attachment, which represents an equally important issue for aquaculture operations, and further studies are needed [51,53]. Furthermore, one of the main factors influencing the survival of mussels exposed to various chemical treatments, besides size class, is valve-gaping, which determines the level of soft tissue exposure [29,33,45]. Valves are opened most of the time in raft-cultivated mussels, following mainly a circadian rhythmicity rather than a tidal one [54]. Therefore, it is crucial to ensure valve closure in most individuals prior to treatment exposure, using methods like shaking or freshwater immersion. In our study, all treatments were applied to individuals that were previously emerged for at least 2–3 min, thus promoting immediate valve closure [29,33,47]. However, the mortality obtained in the sodium hypochlorite and acetic acid groups likely indicates some level of contact between chemicals and soft tissues, as a result of gaping mussels. In addition, not only do declustered mussels present increased gaping and byssus production, but some individuals may have imperfect valve sealings as a result of variability in shell shape, enhancing the chemical exposure [29,53]. Interestingly, gapping may be more preponderant in higher size classes of mussels, which may represent another added advantage for applying these methods in earlier life stages, including prophylactic treatments [33]. In mussel seed, invasive tunicates are also likely to be in earlier development stages, with lower adhesion surface, and thus are more vulnerable to treatments, particularly mechanical removal. The length of the present trial was probably too short to detect possible differences in the mussels’ SGR. Andrade et al. [40] also did not obtain differences in condition index of *M. galloprovincialis* submitted to air exposure at increased temperatures. However, some treatments may impair physiological performance in the long term, as reported by Thompson et al. [55], who obtained lower growth rates in *M. edulis* exposed to chlorination, and further studies are needed.

## 5. Conclusions

Our results highlight the efficacy of the acetic acid treatment for eradicating the ascidian *S. plicata*. However, the method still needs further research to minimize mussel mortality. On the other hand, exposure to air, freshwater and hypersaline solution did not induce significant mortality in tunicates. These treatments may need longer exposures to achieve better results. Furthermore, the duration of the present trial was too short to detect differences in mussel and tunicate growth. Therefore, further studies are needed to investigate the long-term effect of the different treatments on mussel growth, as well as the susceptibility of tunicates to treatments according to sexual maturation. Our results should also be interpreted in the context of laboratory conditions, namely the sample size. The application of treatments to declustered mussels may have favored a higher mortality; thus, the same procedures applied in the field are expected to produce different results. Other factors such as the species cultured, culture technique and the fouling community may also affect treatments success [7]. The treatments are likely to produce more effective results than prophylactic measures, applied in a controlled environment in mussel seed, thereby further inhibiting tunicate proliferation. Moreover, because *S. plicata* is a food product in Korea and the Mediterranean [56], this invasive species can represent a commercial opportunity.

## Figures and Tables

**Figure 1 animals-13-01541-f001:**
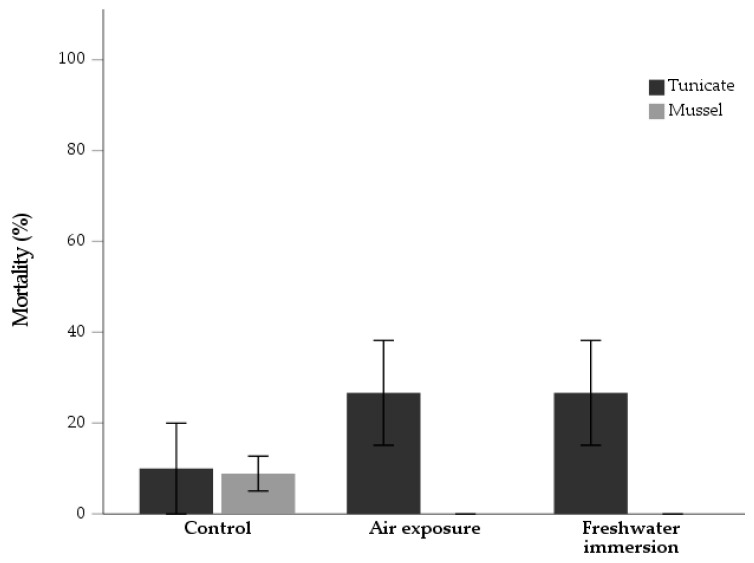
Mortality rate (%) (*n* = 3) of the tunicate *Styela plicata* and the mussel *Mytilus edulis* at the end of a 30-day rearing period following the application of two experimental treatments (air exposure and freshwater immersion). Experiments were run in triplicate, with a control group.

**Figure 2 animals-13-01541-f002:**
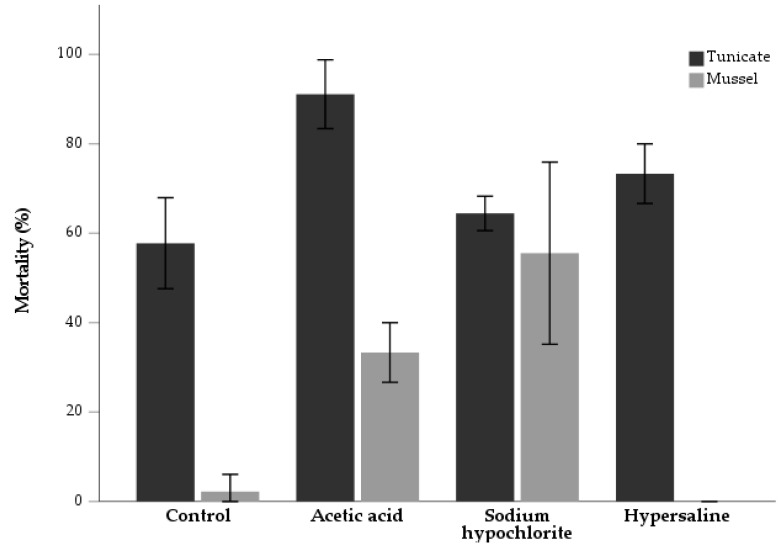
Mortality rate (%) (*n* = 3) of tunicate *Styela plicata* and mussel *Mytilus edulis* at the end of a 15-day rearing period following the application of three experimental treatments (immersions in acetic acid, sodium hypochlorite and hypersaline solutions). Experiments were run in triplicate with a control group.

**Figure 3 animals-13-01541-f003:**
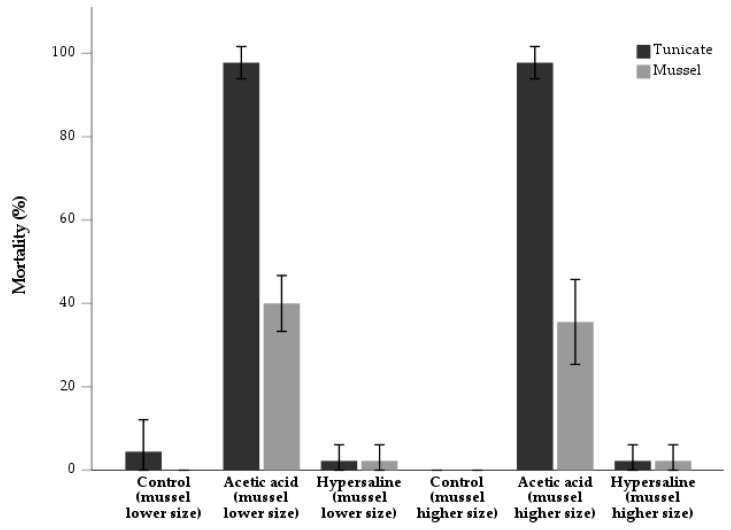
Mortality rate (%) (*n* = 3) of tunicate *Styela plicata* and two size classes of mussel *Mytilus edulis* at the end of a 15-day rearing period following the application of two experimental treatments (immersions in acetic acid and hypersaline solutions). Experiments were run in triplicate with a control group.

**Figure 4 animals-13-01541-f004:**
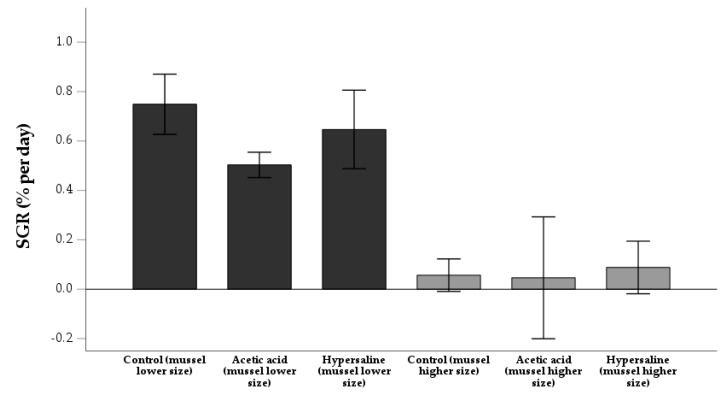
Specific growth rate (SGR) (% day^−1^) (*n* = 3) of two size classes of mussel *Mytilus edulis* at the end of a 15-day rearing period following the application of two experimental treatments (immersions in acetic acid and hypersaline solutions). Experiments were run in triplicate with a control group.

**Table 1 animals-13-01541-t001:** Summary of the experimental treatments tested in this study. The exposure times were applied simultaneously to both the mussels and tunicates.

**Trial 1** (30 Days)	
Air exposure	6 h (repeated 15 days later)
Freshwater immersion	30 min (repeated 15 days later for 1 h)
**Trial 2** (15 days)	
Acetic acid	1 min
Sodium hypochlorite	1.5 min
Hypersaline solution	20 s
**Trial 3** (15 days)	
Acetic acid Hypersaline solution	1 min
	20 s

**Table 5 animals-13-01541-t005:** Maturity index of the tunicate *Styela plicata* at the beginning (T0) and end (T1) of a 15-day period, reared jointly with two size classes of mussel *Mytilus edulis*, in which both species were previously submitted to two experimental treatments (acetic acid, hypersaline solution). Experiments were run in triplicate with a control group.

	Maturity Index	Mussel Lower Size Class Group			Mussel Higher Size Class Group		
		Control	Acetic Acid	Hypersaline Solution	Control	Acetic Acid	Hypersaline Solution
Oocytes (T0)	I	7.78%					
	II	42.67%					
	III	49.56%					
Oocytes (T1)	I	18.97%	53.33%	16.00%	14.05%	50.00%	14.87%
	II	70.26%	46.67%	68.67%	63.57%	43.33%	72.05%
	III	10.77%		15.33%	22.38%	6.67%	13.08%
Male follicles (T0)	I						
	II	86.67%					
	III	13.33%					
Male follicles (T1)	I				28.57%		46.15%
	II	61.54%	100%	73.33%	57.14%		30.77%
	III	38.46%		26.67%	14.29%	100%	23.08%

## Data Availability

The datasets generated during and/or analysed for the current study may be available from the corresponding author on reasonable request.

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
