# Peer review of "Comparison of the Efficiency of Different Eradication Treatments to Minimize the Impacts Caused by the Invasive Tunicate Styela plicata in Mussel Aquaculture"

_animals, 2023, doi:10.3390/ani13091541_

Round 1

Reviewer 2 Report

Given the serious effects resulted from biological invasions, research in tunicate Styela plicata eradication treatments is necessary, especially to mussel aquaculture in EU. In this sense, the work is interesting and a potentially valuable contribution. However, this is an early paper and much work remains before the findings could be used in the practice. Even so, the work is worthy of encouragement.

Introduction

I think the amount increasing course of Styela plicata in Albufeira lagoon should be introduced. Its biological habit and environmental preference should be taken into account when deciding the research programme.

Materials and Methods Results

In Animal collection and maintenance, how about the mortality of tunicate and mussels? And how many days? These are a useful guideline when evaluating the results in the following trials. Besides, only 10 or 15 individuals in a single tank seems inadequate. This would reduce the credibility of evaluation results.

In Trial 1, I think 6h of air exposure is not enough, even though this is the average period of emersion during low tide.

In Trial 2, how did the author decide the treatment intensity, and duration ? Theoretically, the intensity and duration needs to be explored before this work, like concentration gradient.

In Trial 3, still, the sample quantities are not inadequate, leading to curious and abnormal results. For example, the mortality rates of tunicates disposed by acetic acid and hypersaline in both mussels’ lower size and higher size are exactly the same.

Discussion

Subheadings are needed. Please reorganize this part.

Most of this part discussed the performance of other species. However, as the authors mentioned, different results are inevitable between species. As a result, the basic biology of S. plicata is the key point .

Line 329-331 Is this another experiment on a published paper? If not, I think it should not be used here as evidence.

Conclusions

Seems verbose. Please focus on the findings and significance of this study.

References

All references format should be standardized. For example, to italicize all scientific names.

Round 2

Reviewer 1 Report

The authors have responses to my comments adequately and they have improve the manuscript substantially. I have no further comments.

Author Response

The authors are very grateful for the comments and the time the reviewer dedicated to this article. All comments were covered in the previous revision.

Reviewer 2 Report

I think this manuscript could be accepted before little mistakes fixed. For example, delete the brackets in line31 and line93; line50 "1,108"; line53 delete","; line67 the sentence should be reorganized.  In addition, line93-95, if this is an important part of the paper, the result should be concluded in Abstract. 

Author Response

The authors are very grateful for the comments and the time the reviewer spent on this article. These minor corrections and corrections have been addressed in this new version of the article.